# Recursive Context Propagation Network for Semantic Scene Labeling

**Abhishek Sharma**
University of Maryland
College Park, MD
bhokaal@cs.umd.edu

**Oncel Tuzel**          **Ming-Yu Liu**
Mitsubishi Electric Research Labs (MERL)
Cambridge, MA
{oncel,mliu}@merl.com

## Abstract

We propose a deep feed-forward neural network architecture for pixel-wise semantic scene labeling. It uses a novel recursive neural network architecture for context propagation, referred to as rCPN. It first maps the local visual features into a semantic space followed by a bottom-up aggregation of local information into a global representation of the entire image. Then a top-down propagation of the aggregated information takes place that enhances the contextual information of each local feature. Therefore, the information from every location in the image is propagated to every other location. Experimental results on Stanford background and SIFT Flow datasets show that the proposed method outperforms previous approaches. It is also orders of magnitude faster than previous methods and takes only 0.07 seconds on a GPU for pixel-wise labeling of a $256 \times 256$ image starting from raw RGB pixel values, given the super-pixel mask that takes an additional 0.3 seconds using an off-the-shelf implementation.

## 1  Introduction

Semantic labeling aims at getting pixel-wise dense labeling of an image in terms of semantic concepts such as tree, road, sky, water, foreground objects etc. Mathematically, the problem can be framed as a mapping from a set of nodes arranged on a 2D grid (pixels) to the semantic categories. Typically, this task is broken down into two steps - feature extraction and inference. Feature extraction involves retrieving descriptive information useful for semantic labeling under varying illumination and view-point conditions. These features are generally color, texture or gradient based and extracted from a local patch around each pixel. Inference step consists of predicting the labels of the pixels using the extracted features. The rich diversity in the appearance of even simple concepts (sky, water, grass) makes the semantic labeling very challenging. Surprisingly, human performance is almost close to perfect on this task. This striking difference of performance has been a heated field of research in vision community. Past experiences and recent research [1, 2, 3] have conclusively established that the ability of humans to utilize the information from the entire image is the main reason behind the large performance gap. Interestingly, [2, 3] have shown that human performance in labeling a small local region (super-pixel) is worse than a computer when both are looking at only that region of the image. Motivated from these observations, increasingly sophisticated inference algorithms have been developed to utilize the information from the entire image. Conditional Random Fields (CRFs) [4] and Structured Support Vector Machines (SVMs) [5] are among the most successful and widely used algorithms for inference.

We model the semantic labeling task as a mapping from the set of all pixels in an image $\mathbf{I}$ to the corresponding label image $\mathbf{Y}$. We have several design considerations: (1) the mapping should be evaluated fast, (2) it should utilize the entire image such that every location influences the labeling of every other location, (3) mapping parameters should be learned from the training data, (4) it should scale to different image sizes. In addition, good generalization requires limiting the capacity of

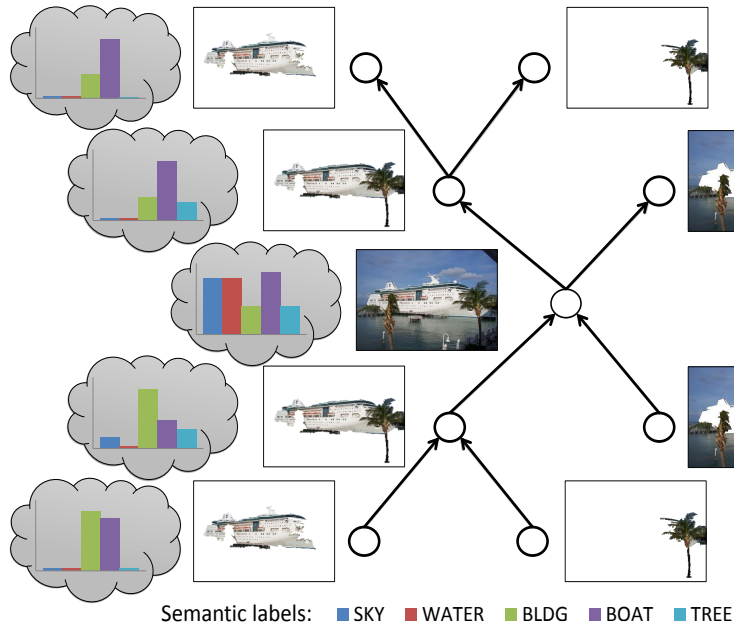

Semantic labels: ■ SKY ■ WATER ■ BLDG ■ BOAT ■ TREE

Figure 1: Conceptual illustration of recursive context propagation network (rCPN). rCPN recursively aggregates contextual information from local neighborhoods to the entire image and then disseminates global context information back to individual local features. In this example, starting from confusion between boat and building, the propagated context information helps resolve the confusion by using the feature of the water segment.

the mapping while still utilizing the entire image information at once. For example, a simple fully-connected-linear mapping from $\mathbf{I}$ to $\mathbf{Y}$ requires 4 Trillion parameters for an image of size $256 \times 256$, but it will fail to generalize under practical conditions of limited training data.

Considering the requirements discussed above, we designed the mapping as a single feed-forward neural network with carefully controlled capacity by parameter sharing. All the network parameters are learned from the data and the feed-forward structure allows fast inference. The proposed network can be functionally partitioned into two sub-networks: local feature extraction and recursive context propagation.

As the name implies, local-feature extraction refers to the extraction of pixel- or region-wise visual features for semantic labeling. We used the multi scale convolutional neural network (Multi-CNN) architecture proposed in [6] to get pixel-wise features. Convolutional structure with shared parameters brings down the number of parameters for local feature extraction.

We propose a novel recursive context propagation network (rCPN), which, starting from the local features, recursively aggregates contextual information from local neighborhoods up to the entire image and then disseminates the aggregated information back to individual local features for better semantic classification. rCPN is a recursive neural network with shared parameters through the parse tree hierarchy. A conceptual illustration of this network is given in Figure 1. The scene consists of three segments corresponding to a boat, a tree and a water/sky region. The nodes of the graph (formed by a binary parse tree and its inversion) represent semantic description of the segments. The distributions on the left are probable label distributions for the adjacent segments based on their appearance. Initially (at the bottom), the boat can be confused as a white building, while looking only at the bottom-left segment. The rCPN recursively combines two segment descriptions and produces the semantic description of the combined segment. For example, as the tree is combined with the boat, the belief that the combined segment includes a building increased since usually they appear together in the images. Similarly, after we merge the water/sky segment description with this segment description, the probability of the boat increased since the simultaneous occurrence of water and building is rare. The middle node in the graph (root node of the segmentation tree)

corresponds to the semantic description of the entire image. After all the segment descriptions are merged into a single holistic description of the entire image, this information is propagated to the local regions. It is achieved by recursive updates of the semantic descriptions of the segments given the descriptions of their parent segments. Finally, *contextually enhanced* descriptions of the leaf nodes are used to label the segments. Note that, rCPN propagates segment semantic descriptions but not the label distributions shown in the illustration.

Our work is influenced by Socher et al.'s work [7] that learns a non-linear mapping from feature space to a semantic space, termed as semantic mapping. It is learned by optimizing a structure prediction cost on the ground-truth parse trees of training images or sentences. Next, a classifier is learned on the semantic mappings of the individual local features from the training images. At test time, local features are projected to the semantic space using the learned semantic mapping followed by classification. Therefore, only the information contained in each individual local feature is used for labeling. In contrast, we use recursive bottom-top-bottom paths on randomly generated parse trees to propagate contextual information from local regions to all other regions in the image. Therefore, our approach uses entire image information for labeling each local region. Please see experiments section for detailed comparison.

The main contributions of the proposed approach are:

- The proposed model is scalable. It is a combination of a CNN and a recursive neural network which is trained without using any human-designed features. In addition, convolution and recursive structure allows scaling to arbitrary image sizes while still utilizing the entire image content at once.

- We achieved state-of-the-art labeling accuracy on two important benchmarks while being an order of magnitude faster than the existing methods due to feed-forward operations. It takes only **0.07** seconds on GPU and **0.8** seconds on CPU for pixel-wise semantic labeling of a $256 \times 256$ image, with a given super-segmentation mask, that can be computed using an off-the-shelf algorithm within 0.3 second.

- Proposed rCPN module can be used in conjunction with pre-computed features to propagate context information through the structure of an image (see experiments section) and potentially for other structured prediction tasks.

## 2   Semantic labeling architecture

In this section we describe our semantic labeling architecture and discuss the design choices for practical considerations. An illustration of this architecture is shown in Figure 2. The input image is fed to a CNN, $F_{CNN}$, which extracts local features per pixel. We then use a super-pixel tessellation of the input image and average pool the local features within the same super-pixel. Following, we use the proposed rCPN to recursively propagate the local information throughout the image using a parse tree hierarchy and finally label the super-pixels.

### 2.1   Local feature extraction

We used the multi scale CNN architecture proposed in Farabet et al. [6] for extracting per pixel local features. This network has three convolutional stages which are organized as $8 \times 8 \times 16\ conv \rightarrow 2 \times 2\ maxpool \rightarrow 7 \times 7 \times 64\ conv \rightarrow 2 \times 2\ maxpool \rightarrow 7 \times 7 \times 256\ conv$ configuration, each max-pooling is non-overlapping. After each convolution we apply a rectified linear (ReLU) nonlinearity. Unlike [6], we do not preprocess the input raw RGB images other than scaling it between 0 to 1, and centering by subtracting 0.5. Tied filters are applied separately at three scales of the Gaussian pyramid. The final feature maps at lower scales are spatially scaled up to the size of the feature map at the highest scale and concatenated to get $256 \times 3 = 768$ dimensional features per pixel. The obtained pixel features are fed to a Softmax classifier for final classification. Please refer to [6] for more details. After training, we drop the final Softmax layer and use the 768 dimensional features as local features.

Note that the 768 dimensional concatenated output feature map is still 1/4th of the height and width of the input image due to the max-pooling operations. To obtain the input size per pixel feature map we either (1) shift the input image by one pixel on a $4 \times 4$ grid to get 16 output feature maps that are

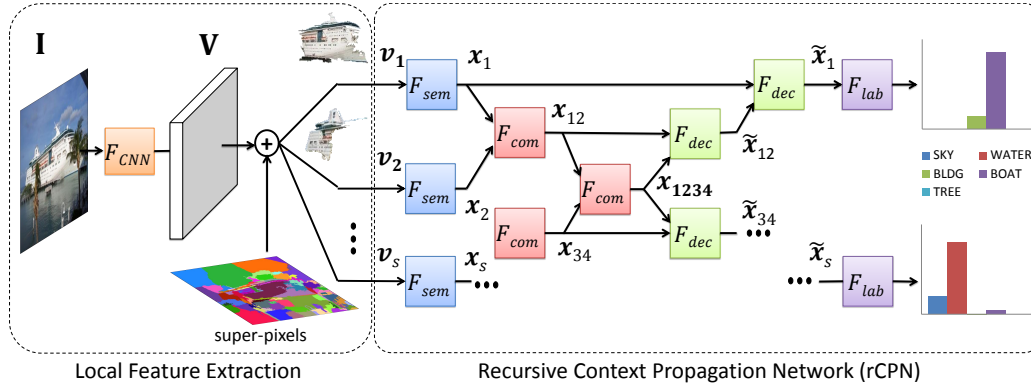

Figure 2: Overview of semantic scene labeling architecture

combined to get the full-resolution image, or (2) scale-up each feature map by a factor of 4 using bilinear interpolation. We refer to the later strategy as *fast* feature map generation in experiments section.

**Super-pixel representation:** Although it is possible to do per pixel classification using the rCPN, learning such a model is computationally intensive and the resulting network is too deep to propagate the gradients efficiently due to recursion. To reduce the complexity, we utilize a super-pixel segmentation algorithm [8], which provides the desired number of super-pixels per image. This algorithm uses pairwise color similarity together with an entropy rate criteria to produce homogenous super-pixels with roughly equal sizes. We average pool the local features within the same super-pixel and retrieve $s$ local features, $\{\mathbf{v}_i\}_{i=1...s}$, one per super-pixel. In our experiments we used $s = 100$ super-pixels per image.

## 2.2 Recursive context propagation network

rCPN consists of four neural networks: $F_{sem}$ maps local features to the semantic space in which the local information is propagated to other segments; $F_{com}$ recursively aggregates local information from smaller segments to larger segments through a parse tree hierarchy to capture a holistic description of the image; $F_{dec}$ recursively disseminates the holistic description to smaller segments using the same parse tree; and $F_{lab}$ classifies the super-pixels utilizing the contextually enhanced features.

**Parse tree synthesis:** Both for training and inference, the binary parse trees that are used for propagating information through the network are synthesized at random. We used a simple agglomerative algorithm to synthesize the trees by combining sub-trees (starting from a single node) according to the neighborhood information. To reduce the complexity and avoid degenerate solutions, the synthesis algorithm favors roughly balanced parse trees by greedily selecting sub-trees with smaller heights at random. Note that, we use parse trees only as a tool to propagate the contextual information throughout the image. Therefore, we are not limited to the parse trees that represent an accurate hierarchical segmentation of the image unlike [9, 7].

**Semantic mapping network** is a feed-forward neural network which maps the local features to the $d_{sem}$ dimensional semantic vector space

$$\mathbf{x}_i = F_{sem}(\mathbf{v}_i; \boldsymbol{\theta}_{sem}), \tag{1}$$

where $\boldsymbol{\theta}_{sem}$ is the model parameter. The aim of the semantic features is to capture a joint representation of the local features and the context, and being able to propagate this information through a parse tree hierarchy to other super-pixels.

**Combiner network** is a recursive neural network which recursively maps the semantic features of two child nodes (super-pixels) in the parse tree to obtain the semantic feature of the parent node (combination of the two child nodes)

$$\mathbf{x}_{i,j} = F_{com}([\mathbf{x}_i, \mathbf{x}_j]; \boldsymbol{\theta}_{com}). \tag{2}$$

Intuitively, combiner network attempts to aggregate the semantic content of the children nodes such that the parent node becomes representative of its children. The information is recursively aggregated bottom-up from super-pixels to the root node through the parse tree. The semantic features of the root node correspond to the holistic description of the entire image.

**Decombiner network** is a recursive neural network which recursively disseminates the context information from the parent nodes to the children nodes throughout the parse tree hierarchy. This network maps the semantic features of the child node and its parent to the contextually enhanced feature of the child node

$$\tilde{\mathbf{x}}_i = F_{dec}([\tilde{\mathbf{x}}_{i,j}, \mathbf{x}_i]; \boldsymbol{\theta}_{dec}). \tag{3}$$

Since we start from the root feature of the entire image and apply the decombiner network top-down recursively until we reach the super-pixel features, every super-pixel feature contains the contextual information aggregated from the entire image. Therefore, it is influenced by every other super-pixel in the image.

**Labeler network** is the final feed forward network which maps the contextually enhanced semantic features ($\tilde{\mathbf{x}}_i$) of each super-pixel to one of the semantic category labels

$$\mathbf{y}_j = F_{lab}(\tilde{\mathbf{x}}_i; \boldsymbol{\theta}_{lab}). \tag{4}$$

Contextually enhanced features contain both local and global context information, thereby leading to better classification.

**Side information:** It is possible to input information to the recursive networks not only at the leaf nodes but also at any level of the parse tree. The side information can encode the static knowledge about the parse tree nodes and is not recurred through the tree. In our implementation we used average $x$ and $y$ locations of the nodes and their sizes as the side information.

## 3 Learning

Proposed labeling architecture is a feed-forward neural network that can be trained end-to-end. However, the recursion makes the depth of the neural network too deep for an efficient joint training. Therefore, we first learn the CNN parameters ($\boldsymbol{\theta}_{CNN}$) using the raw image and the ground truth per pixel labels. The trained CNN model is used to extract super-pixel features followed by training of rCPN ($\boldsymbol{\theta}_{rCPN} = [\boldsymbol{\theta}_{sem}, \boldsymbol{\theta}_{com}, \boldsymbol{\theta}_{dec}, \boldsymbol{\theta}_{lab}]$) to predict the ground truth super-pixel labels.

Feature extractor CNN is trained on a GPU using a publicly available implementation CAFFE [10]. In order to avoid over-fitting we used data augmentation and dropout. All the training images were flipped horizontally to get twice the original images. We used dropout in the last layer with dropout ratio equal to 0.5. Standard back-propagation for CNN is used with stochastic gradient descent update scheme on mini-batches of 6 images, with weight decay ($\lambda = 5 \times 10^{-5}$) and momentum ($\mu = 0.9$). It typically took 6-8 hours of training on a GPU as compared to 3-5 days training on a CPU as reported in [6]. We found that simply using RGB images with ReLU units and dropout gave slightly better pixel-wise accuracy as compared to [6].

rCPN parameters are trained using back-propagation through structure [11], which back-propagates the error through the parse tree, from $F_{lab}$ to $F_{sem}$. The basic idea is to split the error message at each node and propagate it to the children nodes. Limited memory BFGS [12] with line-search is used for parameter updates using publicly available implementation[1]. From each super-pixel we obtained 5 different features by average pooling a random subset of pixels within the super-pixel (as opposed to average pooling all the pixels), and used a different random parse tree for each set of random feature, thus we increased our training data by a factor of 5. It typically took 600 to 1000 iterations for complete training.

## 4 Experiments

We extensively tested the proposed model on two widely used datasets for semantic scene labeling - Stanford background [13] and SIFT Flow [14]. Stanford background dataset contains 715 color images of outdoor scenes, it has 8 classes and the images are approximately $240 \times 320$ pixels.

We used the 572 train and 143 test image split provided by [7] for reporting the results. SIFT Flow contains 2688, $256 \times 256$ color images with 33 semantic classes. We experimented with the train/test (2488/200) split provided by the authors of [15]. We have used three evaluation metrics - **Per pixel accuracy (PPA):** ratio of the correct pixels to the total pixels in the test images; **Mean class accuracy (MCA):** mean of the category-wise pixel accuracy; **Time per image (Time):** time required to label an input image starting from the raw image input, we report our time on both GPU and CPU.

The local feature extraction through Multi-CNN [6] encodes contextual information due to large field of view (FOV); the FOV for 1, 1/2 and 1/4 scaled input images is $47 \times 47$, $94 \times 94$ and $188 \times 188$ pixels, respectively. Therefore, we designed the experiments under single and multi scale settings to assess rCPN's contribution. Mutli-CNN + rCPN refers to the case where feature maps from all the three scales (1,1/2 and 1/4), $3 \times 256 = 768$ dimensional local feature, for each pixel are used. Single-CNN + rCPN refers to the case where only the 256 feature maps corresponding to the original resolution image are used. Evidently, the amount of contextual information in the local features of Single-CNN is significantly lesser than that of Multi-CNN because of smaller FOV. All the individual modules in rCPN, $F_{sem}, F_{com}, F_{dec}$ and $F_{lab}$, are single layer neural networks with ReLU non-linearity and $d_{sem} = 60$ for all the experiments. We used 20 randomly generated parse trees for each image and used voting for the final super-pixel labels. We did not optimize these hyper-parameters and believe that parameter-tuning can further increase the performance. The baseline is two-layer neural network with 60 hidden neurons classifier with Single-CNN or Multi-CNN features of super-pixels and referred to as *Multi/Single-CNN + Plain NN*.

## 4.1 SIFT Flow dataset

We used 100 super-pixels per image obtained by method of [8]. The result on SIFT Flow database is shown in Table 1. From the comparison it is clear that we outperform all the other previous methods on pixel accuracy while being an order of magnitude faster. Farabet et al. [6] improved the mean class accuracy by training a model based on the balanced class frequency. Since some of the classes in SIFT Flow dataset are under represented, the class accuracies for them are very low. Therefore, following [6], we also trained a balanced rCPN model that puts more weights on the errors for rare classes as compared to the dominant ones, referred to as Multi-CNN + rCPN *balanced*. Smoothed inverse frequency of the pixels of each category is used as the weights. Balanced training helped improve our mean class accuracy from 33.6 % to 48.0 %, which is still slightly worse than [6] (48.0 % vs 50.8 %), but our pixel accuracy is higher (75.5 % vs 72.3 %). Multi-CNN + rCPN performed better than Single-CNN + rCPN and both performed significantly better than Plain NN approaches, because the later approaches do not utilize global contextual information. We also observed that the relative improvement over Plain NN was more with Single-CNN features which uses less context information than that of Multi-CNN.

## 4.2 Stanford background dataset

We used publicly available super-pixels provided by [7] with our CNN based local features to obtain super-pixel features. A comparison of our results with previous approaches on Stanford background database is shown in Table 2. We outperform previous approaches on all the performance metrics. Interestingly, we observe that Single-CNN + rCPN performs better than Multi-CNN + rCPN for pixel accuracy. We believe that it is due to over-fitting on high-dimensional Multi-CNN features and relatively smaller training data size with only 572 images. Once again the improvement due to rCPN over plain NN is more prominent in the case of Single-CNN features.

**Model analysis:** In this section, we analyze the performance of individual components of the proposed model. First, we use rCPN with hand-designed features to evaluate the performance of context model alone, beyond the learned local features using CNN. We utilize the visual features and super-pixels used in semantic mapping and CRF labeling framework [7, 13], and trained our rCPN module. The results are presented in Table 3. We see that rCPN module significantly improves upon the existing context models, namely a CRF model used in [13] and semantic space proposed in [7]. In addition, CNN based visual features improve over the hand-designed features.

Next, we analyze the performance of combiner and decombiner networks separately. To evaluate combiner network in isolation, we first obtain the semantic mapping ($\mathbf{x}_i$) of each super-pixel's

Table 1: SIFT Flow result

| Method | PPA | MCA | Time (s) CPU/GPU |
|---|---|---|---|
| Tighe, [15] | 77.0 | 30.1 | 8.4 / NA |
| Liu, [14] | 76.7 | NA | 31 / NA |
| Singh, [16] | 79.2 | 33.8 | 20 / NA |
| Eigen, [17] | 77.1 | 32.5 | 16.6 / NA |
| Farabet, [6] | 78.5 | 29.6 | NA / NA |
| (Balanced), [6] | 72.3 | **50.8** | NA / NA |
| Tighe, [18] | 78.6 | 39.2 | $\geq$ 8.4 / NA |
| Pinheiro, [19] | 77.7 | 29.8 | NA / NA |
| Single-CNN + Plain NN | 72.8 | 25.5 | 5.1/0.5 |
| Multi-CNN + Plain NN | 76.3 | 32.1 | 13.1/1.4 |
| Single-CNN + rCPN | 77.2 | 25.5 | 5.1/0.5 |
| Multi-CNN + rCPN | **79.6** | 33.6 | 13.1/1.4 |
| Multi-CNN + rCPN *Balanced* | 75.5 | 48.0 | 13.1/1.4 |
| Multi-CNN + rCPN *Fast* | 79.5 | 33.4 | **1.1/0.37** |

Table 2: Stanford background result

| Method | PPA | MCA | Time (s) CPU/GPU |
|---|---|---|---|
| Gould, [13] | 76.4 | NA | 30 to 600 / NA |
| Munoz, [20] | 76.9 | NA | 12 / NA |
| Tighe, [15] | 77.5 | NA | 4 / NA |
| Kumar, [21] | 79.4 | NA | $\leq$ 600 / NA |
| Socher, [7] | 78.1 | NA | NA / NA |
| Lempitzky, [9] | **81.9** | 72.4 | $\geq$ 60 / NA |
| Singh, [16] | 74.1 | 62.2 | 20 / NA |
| Farabet, [6] | 81.4 | 76.0 | 60.5 / NA |
| Eigen, [17] | 75.3 | 66.5 | 16.6 / NA |
| Pinheiro, [19] | 80.2 | 69.9 | 10.6 / NA |
| Single-CNN + Plain NN | 80.1 | 69.7 | 5.1/0.5 |
| Multi-CNN + Plain NN | 80.9 | 74.4 | 13.1/1.4 |
| Single-CNN + rCPN | **81.9** | 73.6 | 5.1/0.5 |
| Multi-CNN + rCPN | 81.0 | **78.8** | 13.1/1.4 |
| Multi-CNN + rCPN *Fast* | 80.9 | **78.8** | **1.1/0.37** |

Table 3: Stanford hand-designed local feature

| Method | 2-layer NN [7] | CRF [13] | Semantic space [7] | proposed rCPN |
|---|---|---|---|---|
| PPA | 76.1 | 76.4 | 78.1 | 81.4 |

visual feature using rCPN's $F_{sem}$ and append to it the root feature of the entire image to obtain $\mathbf{x}_i^{com} = [\mathbf{x}_i, \mathbf{x}_{root}]$. Then we train a separate Softmax classifier on $\mathbf{x}_i^{com}$. This resulted in better performance for both Single-scale (PPA: 80.4 & MCA: 71.5) and Multi-scale (PPA: 80.8 & MCA: 79.1) CNN feature settings over (Single/Multi)-CNN + Plain NN. As was previously shown in Table 2, decombiner network further improves this model.

**Computation speed:** Our *fast* method (Section 2.1) takes only 0.37 seconds (0.3 for super-pixel segmentation, 0.06 for feature extraction and 0.01 for rCPN and labeling) to label a $256 \times 256$ image starting from the raw RGB image on a GTX Titan GPU and 1.1 seconds on a Intel core i7 CPU. In both of the experiments the performance loss is negligible using the fast method. Interestingly, the time bottleneck of our approach on a GPU is the super-pixel segmentation time.

Several typical labeling results on Stanford background dataset using the proposed semantic scene labeling algorithm are shown in Figure 3.

## 5   Related Work

Scene labeling has two broad categories of approaches - non-parametric and model-based. Recently, many non-parametric approaches for natural scene parsing have been proposed [15, 14, 16, 17, 18]. The underlying theme is to find similar looking images to the query image from a database of pixel-wise labeled images, followed by super-pixel label transfer from the retrieved images to the query image. Finally, a structured prediction model such as CRF is used to integrate contextual information to obtain the final labeling. These approaches mainly differ in the retrieval of candidate images or super-pixels, transfer of label from the retrieved candidates to the query image, and the form of the structured prediction model used for final labeling. They are based on nearest-neighbor retrieval that introduces a performance/accuracy tradeoff. The variations present in natural scene images are large and it is very difficult to cover this entire space of variation with a reasonable size database, which limits the accuracy of these methods. On the other extreme, a very large database would require large retrieval-time, which limits the scalability of these methods.

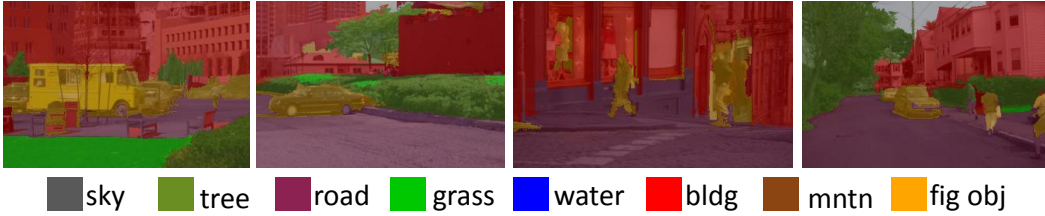

Figure 3: Typical labeling results on Stanford background dataset using our method

Model-based approaches learn the appearance of semantic categories and relations among them using a parametric model. In [13, 20, 2, 3, 22], CRF models are used to combine unary potentials devised through the visual features extracted from super-pixels with the neighborhood constraints. The differences are mainly in terms of the visual features, unary potentials and the structure of the CRF model. Lempitsky et al. [9] have formulated a joint-CRF on multiple levels of an image segmentation hierarchy to achieve better results than a flat-CRF on the image super-pixels only.

Socher et al. [7] learnt a mapping from the visual features to a semantic space followed by a two-layer neural network for classification. Better use of contextual information, with the same super-pixels and features, increased the performance on Stanford background dataset from CRF based method of Gould et al. to semantic mapping of Socher et al. to the proposed work ($76.4\% \rightarrow 78.1\% \rightarrow 81.4\%$). It indicates that neural network based models have the potential to learn more complicated interactions than a CRF. Moreover, they have the advantage of being extremely fast, due to the feed-forward nature. Farabet et al. [6] proposed to learn the visual features from image/label training pairs using a multi-scale convolutional neural network. They reported state-of-the-art results on various datasets using gPb, purity-cover and CRF on top of their learned features. Pinheiro et al. [19] extended their work by feeding in the per-pixel predicted labels using a CNN classifier to the next stage of the same CNN classifier. However, their propagation structure is not adaptive to the image content and only propagating label information did not improve much over the prior work. Similar to these methods, we also make use of the Multi-CNN module to extract local features in our pipeline. However, our novel context propagation network shows that propagating semantic representation bottom-up and top-down using a parse three hierarchy is a more effective way to aggregate global context information. Please see Tables 1 and 2 for a detailed comparison of our method with the methods discussed above.

CRFs model the joint distribution of the output variables given the observations and can include higher order potentials in addition to the unary potentials. Higher order potentials allow these models to represent the dependencies between the output variables, which is important for structured prediction tasks. On the downside, except for a few exceptions such as non-loopy models, inference in these models is NP-Hard that can be only approximately solved and is time consuming. Moreover, parameter learning procedures that are tractable usually limit the form of the potential functions to simple forms such as linear models. In contrast, in our model, we can efficiently learn complex relations between a single output variable and all the observations from an image, allowing a large context to be considered effectively. Additionally, the inference procedure is a simple feed-forward pass that can be performed very fast. However, the form of our function is still a unary term and our model cannot represent higher order output dependencies. Our model can also be used to obtain the unary potential for a structured inference model.

## 6 Conclusion

We introduced a novel deep neural network architecture, which is a combination of a convolutional neural network and recursive neural network, for semantic scene labeling. The key contribution is the recursive context propagation network, which effectively propagates contextual information from one location of the image to other locations in a feed-forward manner. This structure led to the state-of-the-art semantic scene labeling results on Stanford background and SIFT Flow datasets with very fast processing speed. Next we plan to scale-up our model for recently introduced large scale learning task [23].

## Footnotes

[1] http://www.di.ens.fr/~mschmidt/Software/minFunc.html

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
