[Reviews · NeurIPS 2014]

Submitted by Assigned_Reviewer_38

The paper presents a new recursive neural network architecture for semantic scene labeling and shows that it outperforms previous approaches on two standard datasets in terms of pixel accuracy.

The paper is generally very well written and the proposed model seems quite natural and conceptually clean compared to its main competitors.

My only major concern is that the paper doesn’t separately evaluate the effects of the combiner and decombiner networks. An even simpler model could use the combiner network to recursively collapse everything to a single root node (as is done already) but then directly feed the output of the F_sem network along with the root node F_com features into each corresponding F_lab network. It seems like this might be a way of showing that the decombiner network is an important component.

Some minor points:
Line 154 - Is the max pooling overlapping or non-overlapping?

Lines 265-268 are a bit confusing. How does averaging (features of?) 5 pixels give 5 different features for each superpixel?
Summary: A good paper that presents a new a model that is conceptually simple and achieves very good results on two standard scene labeling datasets.

Submitted by Assigned_Reviewer_41

The paper describes a novel approach to the task of pixelwise semantic labeling.

The core of this approach is a recursive neural network architecture that enables context propagation, i.e. each decision to label one particular pixel can benefit from a global representation.

It is a nice contribution to the challenging task of semantic labeling.

# Quality

The paper is nicely written, figures of good quality.

Experiments are nicely described, and I'm glad the authors reported mean-class accuracy (with and without balanced sampling during training).

# Clarity

Quite clear overall, although the description of the proposed recursive method is not completely obvious (it takes 2 or 3 passes to get it 100%).

# Originality

The work presented in this paper builds on two previously published methods: (1) the multi-scale feature extraction scheme proposed in Farabet et al. [6]; and (2) the recursive neural net architecture for semantic mapping, presented in Socher et al. [7].

Its main originally of this work is the recursive context propagation network, which is worth publishing.

The work also proposes a few improvements over [6]: use of GPU, and use of the recently introduced Dropout method, which by itself improves pixelwise accuracy.

# Significance

Full pixelwise semantic labeling is the next frontier in image understanding. It is a complex task that requires a form of advanced reasoning system (graphical model / structured predictor or as proposed in this paper a recursive context propagation system), and an efficient, learnable feature extraction pipeline.

One of the reasons the task of semantic labeling is challenging is the combinatorial aspect of the segmentation: the number of potential segmentation candidates to evaluate is extremely large.

This work is a nice contribution to this problem, and proposes an elegant method to reduce the number of segmentations that need evaluation.
Summary: The paper describes a novel approach to the task of pixelwise semantic labeling. The core of this approach is a recursive neural network architecture that enables context propagation, i.e. each decision to label one particular pixel can benefit from a global representation.

Submitted by Assigned_Reviewer_42

In this paper, the authors propose a model called 'rCPN' composed of a sequence of neural networks for semantic scene labeling (label every pixel in a scene by the class it belongs to). They use the network proposed by Farabet et al. (multiscale CNN, with some small differences such as using ReLU non-linearity and Dropout) to extract features that describe local characteristic of superpixels. These local features
are then passed through a network to map them into a semantic space where local information can be propagated. This local information is then recursively aggregated using a parse tree hierarchy. Once the scene is completely aggregated, a third network is used to disseminate the holistic description back into the initial superpixel segmentation. Finally, a fourth network is used to classify the superpixels into different classes.
The training is done into two separate stages: first the mutliscale CNN is trained from raw pixel. Then the image is segmented into a set of superpixels, and binary parse trees are made. Finally, rCPN (consisting of the four networks) are trained to predict the label of each superpixel.

The paper is well written in general and the ideas are easy to get. The Related Works section could be inserted between Introduction and Learning sections instead. It would be interesting to make a better comparison with other models also using neural networks in the Related Works section (tradeoff between improved results/complexity of model). For the Experiments sections, the Stanford Background dataset results could be introduced before the SIFT Flow dataset ones, as it is a simpler dataset somehow. In line 259, when comparing the training time with Farabet et al you should mention you do GPU training while Farabet et al. explicitly mention no use of GPU. Also, in Table 1, you add the inference time for Pinheiro et al which I couldn't find in the cited paper, maybe you should just leave NA as in the others.

The authors achieve good results with the expense of a complex, non-straightforward model (many different steps during the training).
For the experiments, maybe it would be interesting to mention the other results from Farabet (pure multi-CNN and multi-CNN/superpixel) and Pinheiro (considering smaller output resolution) with faster models, which achieve relatively good results with faster inference time (and a more straightforward models trained from raw data in one step). Also, how many parameters does the model has, compared to other approaches?
Summary: Good results achieved with a model based on two previous works. Even though the work is not very original, the experimental results are good.
Author Feedback
Author rebuttal: We thank reviewers for their encouraging comments and valuable suggestions. We are glad that all the reviewers assessed the contributions of our paper positively. Below we briefly address their questions:

Reviewer 38
Max-pooling is non-overlapping.
Multi-scale convnet gives the local feature for each pixel and we aggregate pixel-wise local features within a super-pixel to get the local feature of each super-pixel. To increase the amount of training data we used bootstrapping: We simulated training data by randomly taking 5 pixels within each super-pixel and averaged them to get the local feature for the super-pixel (rather than averaging over all the pixels within the super-pixel). We repeated the random sampling within each super-pixel 5 times to obtain 5 different training samples for each super-pixel. We will clarify this point in the final version.
Thanks for the suggestion to analyze the combiner – decombiner networks separately. We will comment about this analysis in the final version.

Reviewer 41
Thanks for your positive feedback.

Reviewer 42
Thanks for the suggestion to change the order of the two dataset evaluations (natural ordering is increasing difficulty of datasets). The reason for the current presentation order is to provide an additional experiment on the Stanford dataset evaluating the contribution of rCPN alone (without CNN) using hand designed features that are supplied by the previous work.
Thanks for pointing out the typo that is reporting the time of Pinhero et al.’s method on SIFT flow dataset. We will remove this timing in the final version.
We have only included the best results (in terms of accuracy) from the previous work. We will comment about the faster methods proposed in these works in the final version. Note that the faster methods reported in these papers have significantly lower accuracy than our method with approximately the same running time on CPU.
Our CNN structure is essentially the same as Farabet et al.’s method with ~800K parameters. The overhead of the rCPN, which has ~30K-60K parameters, is relatively insignificant.